# Application of a Novel Quantitative Trait Locus Combination to Improve Grain Shape without Yield Loss in Rice (*Oryza sativa* L. spp. *japonica*)

**DOI:** 10.3390/plants12071513

**Published:** 2023-03-30

**Authors:** Hyun-Su Park, Chang-Min Lee, Man-Kee Baek, O-Young Jeong, Suk-Man Kim

**Affiliations:** 1Crop Breeding Division, National Institute of Crop Science, Rural Development Administration, Wanju 55365, Republic of Korea; 2Department of Ecological & Environmental System, Kyungpook National University, Sangju 37224, Republic of Korea

**Keywords:** QTL, grain shape, *japonica*-type, yield loss, marker-assisted breeding, *Oryza sativa* L.

## Abstract

Grain shape is one of the key factors deciding the yield product and the market value as appearance quality in rice (*Oryza sativa* L.). The grain shape of *japonica* cultivars in Korea is quite monotonous because the selection pressure of rice breeding programs works in consideration of consumer preference. In this study, we identified QTLs associated with grain shape to improve the variety of grain shapes in Korean cultivars. QTL analysis revealed that eight QTLs related to five tested traits were detected on chromosomes 2, 5, and 10. Among them, three QTLs—*qGL2* (33.9% of PEV for grain length), *qGW5* (64.42% for grain width), and *qGT10* (49.2% for grain thickness)—were regarded as the main effect QTLs. Using the three QTLs, an ideal QTL combination (*qGL2^P^* + *qGW5^P^* + *qGT10^B^*) could be constructed on the basis of the accumulated QTL effect without yield loss caused by the change in grain shape in the population. In addition, three promising lines with a slender grain type were selected as a breeding resource with a *japonica* genetic background based on the QTL combination. The application of QTLs detected in this study could improve the grain shape of *japonica* cultivars without any linkage drag or yield loss.

## 1. Introduction

Rice (*Oryza sativa* L.) is a key staple food crop in over 100 countries around the world. However, the risk of yield loss based on climate change, which increases stress factors and decreases arable land and water resources, is increasing every year [1,2]. In addition, considering the growing trend of the population worldwide, the continued increase in rice yield should remain stable in the future [3,4].

In addition to panicle number, spikelet number per panicle, and grain filling rate as the major determinants of grain weight and yield [5], grain shape is characterized by the combination of four parameters (grain length [GL], grain width [GW], the ratio of length to width [RLW], and grain thickness [GT]) and is associated with the quality of grain appearance [6]. Therefore, grain shape is strongly correlated with not only rice yield but also grain appearance, a quality trait that influences market value [7,8]. Rice consumer preferences for grain appearance vary widely depending on the cooking method, dietary habits, and taste preferences. Individuals living in *indica* rice cultivation regions tend to prefer long, slender grains, whereas those living in *japonica* rice cultivation areas, such as Korea, northern China, and Japan, prefer short, round grains [9]. Thus, almost all *japonica* cultivars in Korea have a monotonous short-round shape with an RLW of <2.0 because of biased selection pressure in rice breeding programs based on consumer preference [10]. A monotonous and uniform shape could cause difficulty in rapidly adapting to global rice market change caused by the increase in rice consumption worldwide or variation in rice cultivation environments due to climate change [1,11]. Therefore, exploring the genetic basis of grain shape and identifying novel quantitative trait loci (QTLs) for improving grain yield and quality are necessary.

According to Zuo and Li [12], grain shape is a complex polygenic QTL that is slightly influenced by the environment. Huang et al. [13] reported that improving grain shape is difficult because some QTLs may have a dominant effect on a specific grain trait, whereas most QTLs could affect multiple traits. However, notable achievements associated with rice grain traits have been identified via gene-based cloning, fine-mapping, and primary QTL analysis. Since the identification of *GS3* as a cloned gene with respect to rice grain size, nine genes, including *GS3*, have been reported so far with regard to grain shape, size, or weight. *GS3* encodes a putative transmembrane protein to enhance GL and grain weight using a backcross population of Minghui 63 (large grain) × Chuan 7 (small grain) [14]. *GW2*, which suppresses cell division by controlling proteolysis, was cloned and characterized using QTL, which negatively affected grain weight [15]. In addition, *GW5* and *qSW5* increase GW by regulating cell division during seed development; these genes were identified in a different resource but finally at the same locus [16,17]. *GS5*, encoding a putative serine carboxypeptidase, which increases GW and grain weight, was detected near *GW5*/*qSW5* on chromosome 4 [18]. *GL3.1* influences protein phosphorylation in the spikelet to accelerate cell division and increases GW and grain yield when an allele is derived from a large grain WY3 cultivar [19]. Regarding grain slenderness, *GW8* encodes a positive regulator of cell proliferation (*OsSPL16*), which regulates the formation of a slender grain with a better quality of appearance [20]. *GW7* has the same effect as *GW8* on grain shape when constructing an *OsSPL16-GW7* regulatory module [21] Two genes, *GL2* [22] and *GS2* [23], encoding growth-regulating factor 4 (*OsGRF4*) at the same locus on chromosome 2, which enhanced grain weight and yield by increasing cell size and number, were reported separately using map-based cloning. Recently, Duan et al. (2017) cloned and characterized *GSE5* at the *GW5*/*qSW5* locus on chromosome 5, which has different haplotypes with *GW5*/*qSW5*. The haplotype *GSE5^DEL1^*^+*IN1*^, which leads to the formation of wide and heavy grains, is rarely observed in wide *indica* varieties. Meanwhile, *smg1* [24] and *smg11* [25], which are associated with grain size, were isolated via map-based cloning using mutagenized cultivars. The mutation of genes related to the brassinosteroid biosynthetic pathway decreased grain size by suppressing the regulation of cell expansion.

Moreover, many QTLs were fine-mapped by narrowing down the target regions through the development of molecular technology with regard to GL and grain size. *qGL-6* was anchored in an approximately 1.26-Mb region on chromosome 6, involving *OsARF19* as a candidate gene, thereby increasing GL using an allele derived from Z1392 [26]. Four QTLs, namely *qGL7*, *GS7*, *qSS7*, and *qGRL7*, contributing to an increase in GL, were fine-mapped within 4.8 to 2390 kb on chromosome 7 using populations derived from the cross of *indica* × *japonica*. [9,27,28,29]. QTLs such as *qGRL1*, *qGL4b*, and *qGL-3a*, which are associated with an increase in GL, were also fine-mapped on chromosomes 1, 4, and 4, respectively [29,30,31]. In addition, several QTLs associated with rice grain traits have been identified via primary QTL mapping.

In the present study, we identified QTLs associated with grain shape to improve the diversity of the shape of *japonica* rice cultivars in Korea using recombinant inbred lines (RILs) derived from a cross between *japonica* and tropical *japonica* cultivars. Based on QTL analysis via inclusive composite interval mapping (ICIM), QTL combinations may increase GL or RLW without decreasing grain weight, thereby allowing the comparison of accumulated QTL effects. Therefore, the QTL combination could be used in rice breeding programs to extend the diversity of grain shape in Korean cultivars.

## 2. Results

### 2.1. Evaluation of Tested Traits in the RIL Population

During 2019 and 2020, the main agronomic traits, including days to heading and five traits related to grain shape (GL, GW, GT, RLW, and 1000-grain weight [TGW]), were evaluated. Based on skewness and kurtosis as well as the Shapiro–Wilk test for normality, most traits were normally distributed over the two years (Figure 1).

### 2.2. Correlation Analysis of the Five Traits

To describe the relationships among the five traits, correlation coefficients (*r*) were calculated using the psych package in R (Table 1). RLW indicating the grain shape strongly correlated with GW (*r* = −0.89, *p* < 0.001) (Figure 2). Although GL correlated with RLW (*r* = 0.68 ***), any correlation was not noted with other traits. TGW positively correlated with GW (*r* = 0.83 ***) and GT (*r* = 0.86 ***) at *p* values of <0.001 but was not affected by GL.

### 2.3. QTL Analysis

Genotypic and phenotypic data were combined to analyze QTLs using the RIL population with a *japonica* background. Based on the analysis, eight QTLs related to the tested traits were detected on chromosomes 2, 5, and 10 (Table 2). The threshold LOD score of 3.4 was accepted using 10,000 permutations in IciMapping at *p* values of <0.05. *qGL2*, with an LOD of 8.32, was detected between KJ02_45 and KJ02_47 on chromosome 2, explaining 33.9% of phenotypic variation in ICIM. Regarding GW, two QTLs (*qGW5* and *qGW10*) explained 64.4% and 13.6% of phenotypic variation, respectively, and positively affected the trait of alleles derived from Boramchan. In particular, the locus of *qGW5* was identified as a hot site, which included three QTLs (*qGT5*, *qRW5*, and *qTW5*) anchored between KJ05_13 and KJ05_17 on chromosome 5 (Appendix A). The other loci (*qGW10*, *GT10*, and *qTW10*) were clustered in order within the long arm of chromosome 10.

### 2.4. Effect of QTLs

To verify the effect of the detected QTLs, the phenotypic difference in each related trait was assessed by comparing the mean value of the measured traits in lines classified based on the presence or absence of QTL or QTL combinations (Figure 3). Regarding GL, a single QTL, *qGL2*, was only detected on chromosome 2, and the GL in the group with *qGL2* was significantly longer than that of the group without QTL in the *t*-test (*p* < 0.05). In QTL combinations tested for GW, TGW, and GT, the effects of QTLs were compared using Duncan’s multiple range test (DMRT) at a 5% significance level. The measured values in lines with even one QTL increased remarkably without exception compared with those in lines with no QTL. The increasing effect of accumulated QTLs significantly differed according to the number of QTLs in the case of GT, whereas no difference was observed among the QTLs tested for GW and TG; moreover, no statistical difference was noted among QTLs detected in the same trait despite different LOD.

### 2.5. Determining an Ideal QTL Combination for Grain Shape

To select lines with an increase in RLW without TGW loss, individual lines in the population were classified into eight groups from ty1 to ty8 by combining three QTLs (*qGL2*, *GW5*, and *qGT10*). The mean TGW in each group was analyzed using a one-way analysis of variance, and the differences were compared among the groups using DMRT at a 5% significance level. Regarding TGW, the highest mean values were noted in the ty1 and ty5 groups, whereas the lowest mean values were observed in the ty4 and ty8 groups (Figure 4). The groups with the highest mean values (ty1 and ty5) commonly included *qGT10^B^* and *qGW5^B^* derived from the allele of Boramchan, whereas the groups with the lowest mean values (ty4 and ty8) included alleles derived from Pecos. Regarding RLW, three groups (typ3, ty4, and ty8) showed the highest mean value, and another three groups (ty1, ty5, and ty6) showed the lowest mean value in the analysis. In particular, the ty3, ty4, and ty8 groups commonly had *qGW5^P^* derived from the Pecos allele, whereas the ty1, ty5, and ty6 groups exhibited a specific combination primarily containing *qGW5^B^*.

Meanwhile, the scatter plot showed the distribution of the trend of individual lines based on the type of QTL combination (ty1–ty8) for RLW and TGW (Figure 5). RLW was negatively and strongly correlated with TGW in selected lines (*r* = −0.57, *p* < 0.001). Among lines with a mean TGW of the P1 (Boramchan) level, only three lines in the ty3 group (*qGL2^P^* + *GW5^B^* + *qGT10^B^*) showed a mean RLW of the P2 (Pecos) level. 

### 2.6. Validation Tests for Grain Shape-Related Genes

A haplotype test was performed to validate the presence of previously known genes among the intrinsic genes of the parents used in this study (Table 3). Accordingly, eight specific polymerase chain reaction (PCR)-based DNA markers were used for the validation of eight genes (Appendix A). Polymorphism between parents was confirmed only in *qSW5* among the eight tested genes. Boramchan included an allele of *qsw5_N* and a Pecos-amplified allele of *qSW5*. *qSW5* has three types of alleles—*qsw5_N*, *qsw5_I*, and *qSW5_K*—among which *qsw5-N* and *qsw5_I* are related to increasing GW and GT. Furthermore, *GW7* and *GW8* related to increasing grain weight and yield were confirmed in the parents.

## 3. Discussion

With regard to rice grain shape, the diversity of rice cultivars in Korea is monotonous because of the genetic narrowness of *japonica* rice and consumer preference [11]. This monotony may cause difficulty in responding rapidly to variations in the rice cultivation environment caused by climate change or consumer needs [10]. Therefore, verifying the genetic diversity of grain shape is necessary for the Korean rice breeding program. In this study, 145 RILs obtained from a cross between Boramchan (medium-short grain) and Pecos (medium grain) were used as a mapping population to improve the grain shape of the *japonica*-type rice cultivar in Korea. To obtain phenotypic data regarding grain shape in the population, five traits were evaluated and analyzed. RLW, indicating grain shape, strongly and negatively correlated with GW and GT (Table 1). In particular, GW served as an important factor for determining grain shape in the population. This correlation was consistent with previous studies on grain shape [2,33]. However, GT in this population may be correlated with RLW with a relatively high coefficient of correlation because the grain type of the parents was significantly different.

In QTL analysis, eight QTLs for grain shape were continuously detected during the experimental periods using ICIM analysis on chromosomes 2, 5, and 10 (Table 2). Regarding GL, a novel QTL, *qGL2*, explaining approximately 33.9% of PVE, was delimited within KJ02_45 (24.47 Mb) and KJ02_47 (24.28 Mb) on chromosome 2. QTLs associated with GL were usually alleles derived from *indica* resources. Thus, the use of *qGL2^p^* (allelic to Pecos) derived from *japonica* resources could help improve the GL of *japonica* cultivars without linkage drag caused by a wild cross. Chen et al. (2019) reported that *qGL2^_RF4^* (allelic to *OsGRF4*) displayed a strong effect on GL and a slight effect on GW, and that it overlapped with the locus of *GS2*, which controls grain size at 27.4 Mb on chromosome 2. Meanwhile, in this study, most QTLs associated with grain shape parameters, except for GL, were identified and clustered on chromosomes 5 and 10 (Appendix A). In particular, four QTLs (*qGW5*, *qGT5*, *qRW5*, and *qTW5*) were delimited by the same franking markers, KJ05_13 (4.78 Mb) and KJ05_17 (5.98 Mb), on chromosome 5. *qGW5*, *qGT5*, and *qTW5* were all derived from Boramchan, and only *qRW5* was derived from Pecos. The results of QTLs, which are closely linked to each other, were consistent with the correlation analysis in this study, showing a high degree of correlation among these traits. Regarding grain size, the positions of three genes (*GS5*, *GW5*, and *qSW5*) related to QTLs previously cloned and clustered on chromosome 5 [16,17,18,33] were identified near the QTLs detected in this study. *GS5* encodes a putative serine carboxypeptidase and serves as a positive regulator of grain size. A newly identified QTL [18], *GW5*, controlling rice GW, was cloned; it encodes a novel nuclear protein of 144 amino acids that is localized to the nucleus [16]. *qSW5* is involved in the determination of GW in rice, a significant increase in sink size owing to an increase in cell number in the outer glume of the rice flower [17]. *GW5* and *qSW5*, derived from the parents Asominori and Nipponbare, respectively, were identical at 5.37 Mb of the cloned locus on chromosome 5, whereas *GS5* was located approximately −2 Mb away (3.45 Mb). Considering the target regions detected in this study, including the *GW5*/*qSW5* locus, overlapping of the locus may occur. Three QTLs related to grain weight were detected within 19.47 to 22.49 Mb on chromosome 10. The sum of the phenotypic validation of QTLs was approximately 52.1%, and the alleles derived from Boramchan were associated with the increase in the expression of each target trait. Huang et al. (2013) reported that chromosome 10 was not a key site showing the characteristic distribution of QTLs for grain shape; however, *qGT10* and *qTW10* were identified as novel QTLs within 21.38–22.45 and 21.38–22.49 Mb on chromosome 10, respectively. Meanwhile, the target region of *qGW10* for grain weight was detected within 19.82–21.12 Mb in the region of the major QTL, *TGW10* [34].

To validate the effects of the accumulated QTLs, the phenotypic difference in grain shape was compared in the presence or absence of QTL or QTL combinations (Figure 3). The effect of detected QTLs was assessed by confirming whether increasing the expression of relevant traits was dependent on the number or type of QTLs. *qGL2* can effectively and significantly enhance GL in the tested lines. An accumulated effect was only observed in QTLs (*qGT10* and *qGT5*) associated with GT. Based on these results, three QTLs (*qGL2*, a single-effect QTL; *qGW5*, a QTL with the highest PVE; and *qGT10*, a QTL with an accumulation effect) could be used as selection makers for developing a line with increased GL without grain weight loss. Using the three selected QTLs, we constructed eight QTL combinations and then compared the phenotypic variation in RLW and TWG (Figure 4). When the groups had *qGT10* and *qGW5*, the highest TWG was noted in ty1 and ty5. In groups without *qGW5*, the highest RLW was observed in ty3, ty4, and ty8. Among these groups, ty3, composed of *qGL2^P^* + *qGW5^P^* + *qGT10^B^*, showed a slender grain shape with limited weight loss. The results indicated that *qGW5^B^* is a key factor that increases grain weight while negatively affecting its slender shape. In particular, *qGT10^B^* associated with increasing GT prevented the loss of grain weight and replaced the role of *qGW5^B^* in increasing grain weight in the ty3 group. A total of 13 lines belonged to 3ty, and they were distributed with high RLW and TWG values (Figure 5). Of these lines, five had TGW at the Boramchan level, and three simultaneously showed a grain shape at the Pecos level [35].

Using the parents employed in this study, we identified the allele type of known genes for grain shape using PCR-based DNA markers to detect the genes (Table 3). Among these markers, only the *qSW5*-specific marker showed polymorphism between the parents, and the genes were not validated in the parents, excluding *GW7* and *GW8*. Based on a previous study, approximately 98.6% of *japonica* cultivars in Korea among the tested 286 cultivars had the Nipponbare allele type (*qsw5_N*), enhancing GW and GT, with lines comparable to those of the Kasalath allele type (*qSW5*) [36]. *qGW5^B^* and *qsw5_N* show identical functions in the determination of grain shape; in particular, both may be either multiple alleles or identical genes from different donors considering the target region of *qGW5^B^*, including the possibility of a novel gene.

## 4. Conclusions

To enhance the rice grain shape of *japonica* cultivars in Korea, we identified eight QTLs associated with grain shape in an RIL population with a *japonica* genetic background. The QTLs detected on chromosomes 2, 5, and 10 tended to cluster in the target regions, excluding *qGL2* on chromosome 2. A total of 6 of the QTLs showed major effects on the related traits, with over 20% of PVE. In the analysis, we constructed eight QTL combinations composed of three main effect QTLs and suggested an ideal combination in the ty3 group (*qGL2^P^* + *qGW5^P^* + *qGT10^B^*). In addition, we developed three potential lines (BP27, BP64, and BP139) showing a slender type of *japonica* background without grain weight loss based on the QTL effects (Appendix A). Fine-mapping and functional analysis of the novel QTLs will be performed in a follow-up study along with marker-assisted backcross breeding with selected potential lines. Therefore, the current results can verify the genetic diversity of *japonica*-type cultivars with various grain shapes in rice breeding programs in Korea.

## 5. Materials and Methods

### 5.1. Plant Materials and Mapping Population

The *japonica* high-yield cultivar, Boramchan, and the tropical *japonica* cultivar, Pecos, were crossed to develop a population for investigating rice grain shape. Boramchan is a high-yield rice cultivar with a typical semi-round grain shape and an RLW of 1.61 in the Republic of Korea (Figure 6 and Appendix A). Pecos is a medium-grain cultivar with an RLW of 2.18. To develop a mapping population, 145 RILs (F_7_) derived from a cross between Boramchan and Pecos were established using the single-seed descent method. The RILs were used to construct a linkage map to analyze QTLs associated with the rice grain shape of the tested plants.

### 5.2. Assessments of Traits Associated with Grain Shape

To collect phenotypic data for QTL analysis, five traits related to rice grain shape, namely GL, GW, GT, grain RLW, and TGW, were evaluated for 2 years. Using a caliper, the GL, GW, and GT of well-developed whole grains for each line and brown rice were evaluated with 10 replications, and RLW was calculated based on GL and GW. After measuring the 100-grain weight for each line three times, the mean values were converted into TGW. Rice grain shape was estimated using RLW as follows: short-round shape (bold), <2.0; medium-round shape, 2.01–2.50; long-round shape, 2.51–3.00; slender: >3.01 (RDA 2012) [37].

### 5.3. Genetic Map Construction

The competitive allele-specific PCR (KASP) method, which allows precision biallelic characterization of a single-nucleotide polymorphism (SNP) at specific loci, was used to construct a linkage map and perform QTL analysis of grain shape [38]. KASP revealed 771 marker sets for the whole 12 rice chromosomes, and it was used to construct a genetic map after a parental survey confirmed polymorphisms. Genotypic data were collected based on fluorescence resonance due to allele-specific amplification of a single biallelic SNP. Based on KASP markers showing polymorphism between parents, a genetic linkage map was constructed using QTL IciMapping, version 4.1 [39] 

### 5.4. QTL Mapping

QTL mapping was performed using conventional mapping for ICIM for additive QTLs. For analysis, phenotypic and genotypic data from KASP markers were combined to detect QTLs related to the target traits. Using permutation tests with 1000 replicates (*p* ≤ 0.05), the LOD confirmed the degree of significance of the threshold values.

### 5.5. Data Analyses

A correlation analysis was performed using the “corrplot” package in R version 4.1.0 to analyze the relationship among the traits related to grain shape. The “Scatterplot3d” package was used to construct 3D scatter plots with vertical lines for the grain shape. Significant differences in the mean values were analyzed using DMRT as the post hoc test after the analysis of variance.

### 5.6. Haplotype Test

Eight DNA markers related to genes for grain shape were used to identify the haplotypes of the parents tested in this study (Appendix A). Five genes, *GW2* [15], *qSW5* [17], *TGW6* [32], *GW7* [21], and *GW8* [20], were related to GW, grain thickness, or yield. Three genes or loci, namely *GS3* [14], *qGL3* [19], and *GS5* [18], were related to phenotypic variations in GL, grain weight, and RLW.

## Figures and Tables

**Figure 1 plants-12-01513-f001:**
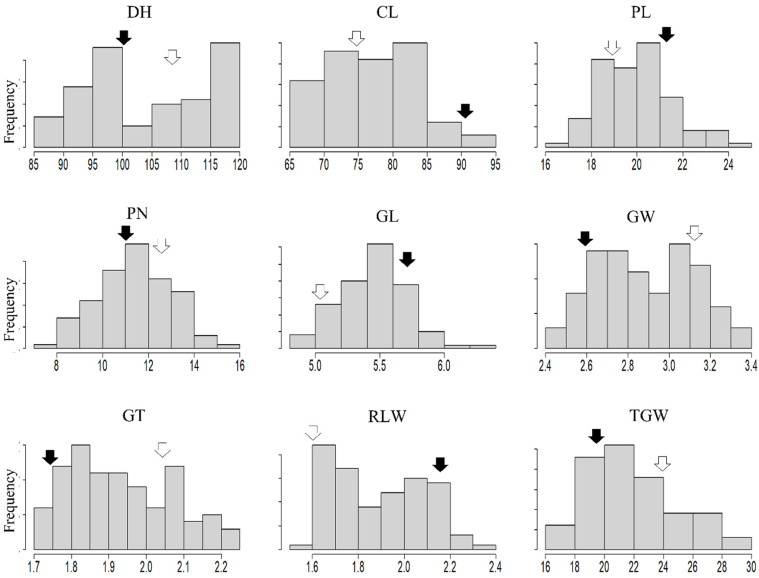
Histograms of the mean values of agronomic traits and traits related to grain shape in the population tested for 2 years. DH, days to heading; CL, culm length; PL, panicle length; PN, panicle number; GL, grain length; GW, grain width; GT, grain thickness; RLW, the ratio of length to width; TGW, 1000 grain weight. The black arrow indicates Pecos, and the white arrow indicates Boramchan.

**Figure 2 plants-12-01513-f002:**
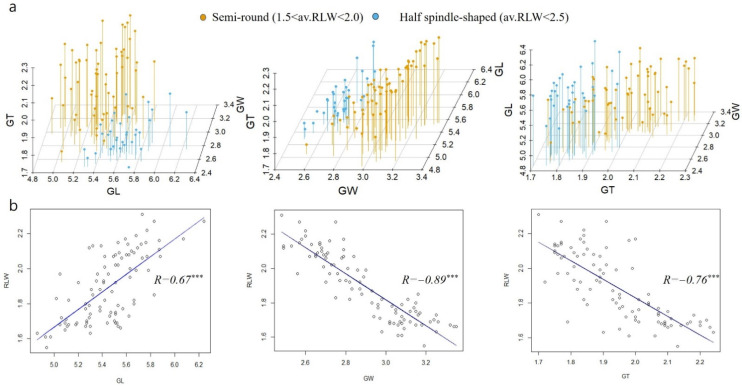
The 3D scatter plots with vertical lines for grain shape, based on the following three traits: grain length (GL), grain width (GW), and grain thickness (GT) (**a**). Scatter plots with a correlation line for the ratio of length to width (RLW) predicted based on the GL, GW, and GT of the population. *r* refers to the correlation coefficient analyzed at *p* < 0.001 (**b**). Significance levels: *** *p* < 0.001.

**Figure 3 plants-12-01513-f003:**
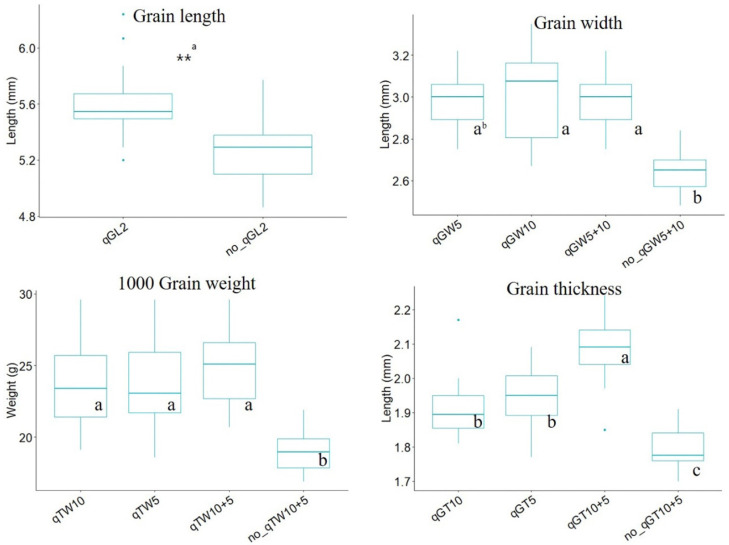
Effect of single QTL and QTL combinations on the following four traits: GL, GW, TGW, and GT. ^a^ Difference between the mean value of each line with *qGL2* by *t*-test. ** indicates significance at *p* < 0.01. ^b^ The letters (a, b, and c) showing difference within the mean values of QTL combination by Duncan multiple range test (DMRT) at a 5% significance level.

**Figure 4 plants-12-01513-f004:**
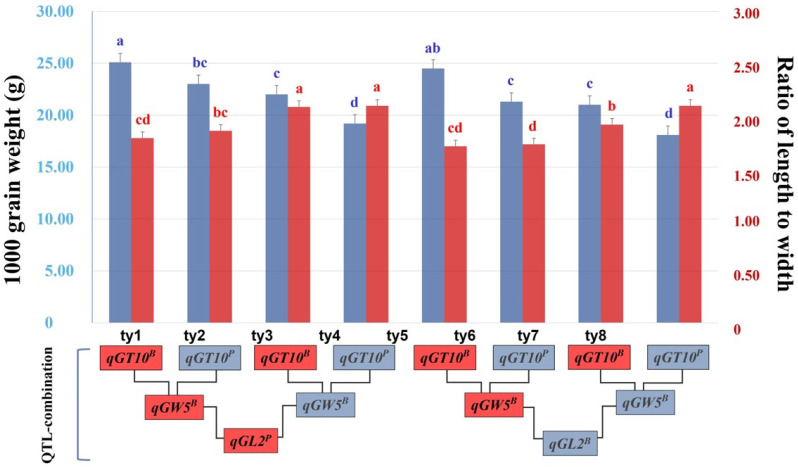
The phenotypic effect of the RLW and TGW by the QTL combination identified in this study. The blue bars and letters on the bar graph indicate grades involved in TWG, and red bars indicate grades involved in RLW. The red boxes of the QTL combination composed of eight groups (ty1–ty8) indicate positive QTL associated with the trait, and the gray boxes indicate negative QTL. The lower-case letters on the error bar indicate significantly different values of measured traits based on the Duncan multiple range test (DMRT) at a 5% significance level.

**Figure 5 plants-12-01513-f005:**
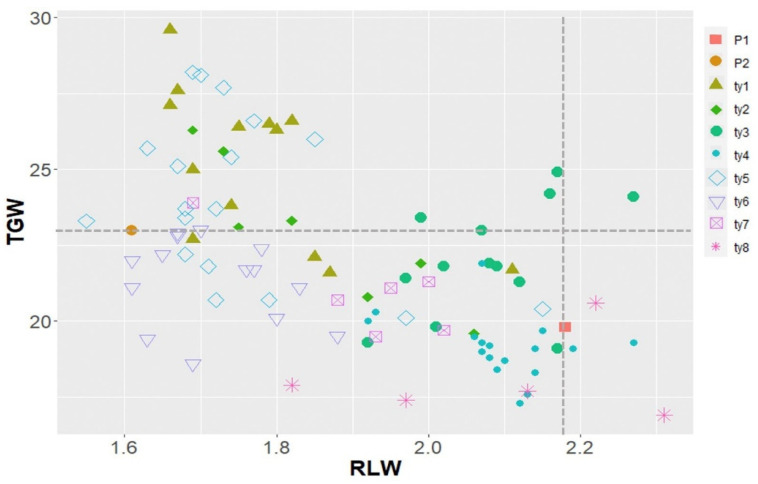
Scatter plot of the ratio of length to width (RLW) and 1000-grain weight (TGW) in 89 lines without heterozygous distribution based on the eight QTL combination types. P1: Boramchan; P2: Pecos; ty1–ty8: eight QTL combinations.

**Figure 6 plants-12-01513-f006:**
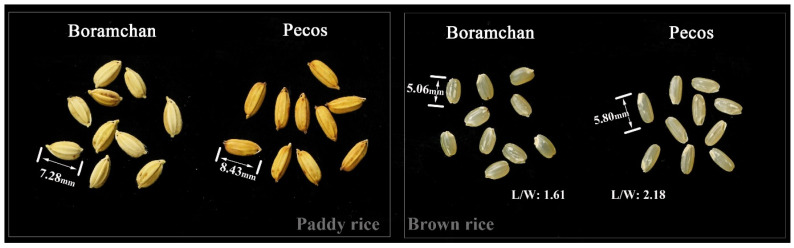
Grain shape of the parents, Boramchan and Pecos, were used in this study. L/W—ratio of length to width.

**Table 1 plants-12-01513-t001:** Correlation of five traits related to grain shape.

	GL	GW	GT	RLW	TGW
GL					
GW	−0.28 **				
GT	−0.24 *	0.87 ***			
RLW	0.68 ***	−0.89 ***	−0.77 ***		
TGW	0.14 ns	0.83 ***	0.86 ***	−0.57 ***	

GL—grain length; GW—grain width; GT—grain thickness; RLW—the ratio of length to width; TGW—1000-grain weight; ns—not significant. Significance levels: * *p* < 0.05, ** *p* < 0.01, and *** *p* < 0.001.

**Table 2 plants-12-01513-t002:** QTL analysis of the traits related to grain shape during 2 years of RIL population.

Traits	QTL	Chr.	Flanking Markers	Mena LOD (Range)	% of Mean PVE	Mean Add
Grain length (GL)	*qGL2*	2	KJ02_45–KJ02_47	8.32 (7.98–8.41)	33.9	−0.15
Grain width (GW)	*qGW5*	5	KJ05_13–KJ05_17	29.32 (283.60–30.04)	64.4	0.17
	*qGW10*	10	KJ10_35–KJ10_39	10.20 (9.88–10.52)	13.6	0.07
Grain thickness (GT)	*qGT5*	5	KJ05_13–KJ05_17	18.64 (18.00–19.23)	49.2	0.09
	*qGT10*	10	KJ10_39–KJ10_41	7.97 (7.66–8.23)	16.5	0.05
Ratio of length to width (RLW)	*qRW5*	5	KJ05_13–KJ05_17	26.77 (21.90–23.64)	72.6	−0.15
1000-grain weight (TGW)	*qTW5*	5	KJ05_13–KJ05_17	15.91 (14.85–16.97)	35.0	1.58
	*qTW10*	10	KJ10_43–KJ10_49	11.00 (10.50–11.50)	22.3	1.26

PVE—percentage of phenotypic variation explained by QTL analysis. Add—additive effect.

**Table 3 plants-12-01513-t003:** Haplotype test for the parents, Pecos and Boramchan, using eight specific DNA markers to characterize genes associated with grain shape.

Gene	Trait	Parents		Allele Type (Effect) ^a^	Reference
		Boramchan (P1)	Pecos (P2)		
*GW2*	Grain width and yield			*gw2^WY3^* (+)	[15]
		*GW2*	*GW2*	*GW2^FAZ1^* (N)	
*GS3*	Grain length and weight			*gs3^Cuuane^* (−)	[14]
				*GS3^hetero-allele^* (M)	
		*GS3*	*GS3*	*GS3^Minghui63^* (+)	
*GL3.1*	Grain size and yield			*qgl3^WY3^* (+)	[19]
		*qGL3*	*qGL3*	*qGL3^FAZ1^* (N)	
*qSW5*	Grain width	*qsw5_N*		*qsw5_N^Nipponbare^* (+)	[17]
				*qsw5_S^SL22^* (M)	
			*qSW5_K*	*qSW5^Kasalath^* (N)	
*GS5*	Grain length and weight	*gs5*	*gs5*	*gs5^H94^* (N)	[18]
				*GS5^Zhensan97^* (+)	
*TGW6*	Grain weight			*tgw6^Kasalath^* (+)	[32]
		*TGW6*	*TGW6*	*TGW6^Nipponbare^* (N)	
*GW7*	Grain width (slenderness)	*gw7*	*gw7*	*gw7^Nipponbare^* (N)	[21]
				*GW7^TFA1^* (+)	
*GW8*	Grain width (slenderness)			*gw8^Basmati385^* (+)	[20]
		*GW8*	*GW8*	*GW8^HJX741^* (N)	

^a^ (+) positive effect increasing relevant traits, (−) negative effect of relevant traits, (M) moderate effect increasing relevant trait, (N) normal grain size.

## Data Availability

Not applicable.

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
