# Peer review of "Application of a Novel Quantitative Trait Locus Combination to Improve Grain Shape without Yield Loss in Rice (Oryza sativa L. spp. japonica)"

_plants, 2023, doi:10.3390/plants12071513_

Round 1
Reviewer 1 Report
Please find my comments in the enclosed file.

Author Response
Response to Reviewer 1 Comments
The following points should clearly be corrected and explained for readers in the manuscript (mn):
Abstract
- L18, scientific name of rice in the first mentioned place should be written with author name such as Oryza sativa L.
Response: That was correctly revised as mentioned by the reviewer
Keywords
- L30, Change Quantitative trait locus to QTL since it was written in title.
Response: In L 31, that was correctly revised as mentioned by the reviewer.
- L30, Oryza sativa can be added.
Response: In L 31, that was correctly revised as mentioned by the reviewer.
Introduction
- L33, please write scientific name of rice with author name.
Response: Line 34 of the revised version of the manuscript (MS) has been correctly revised as suggested by the reviewer
- L35, please check reference style and prepare mn according to journal style.
Response: We used the 'Chicago Manual of Style, 16th edition' as the reference style, which is allowed by this journal as well.
- L35, you can consider to cite the following article related to global warming/climate change, “https://doi.org/10.3390/ agronomy12030557” at the end of sentence.
Response: The current link provided by reviewer does not lead to any information, so we did not reflect it in the revision. We ask for your understanding.
- L38, delete “In addition to”
Response: We believe that the current expression is natural and hope that it will not be modified.
- L56, write full name of QTL in the first mentioned place and then abbreviation can be used.
Response: The full name of QTL was already used before L56 in original MS. You can find it at L57 in the revised version of the MS.
- L61, re-write this sentence. Is GS3 one of nine genes?
Response: The sentence has been revised according to the suggestion of the reviewer in L65.
- Web of Science consist of 52 QTLs articles on grain shape in rice and please mention some of them.
What is the difference your study from these 52 articles?
Response: We differ from a simple QTL identification study, as we reported the successful improvement of grain shape in Japonica rice without yield loss, using the results of QTL analysis as a breeding strategy.
Results
- L112, delete DH, CL, PL, and PN in Figure and Figure caption.
Response: In QTL studies, RILs or F2 populations are commonly used for phenotypic analysis. The reason for presenting the major agricultural trait outcomes of the population used, such as grain shape in this manuscript, is because frequency distribution provides information on whether the selection population is biased for a specific purpose or not, compared to unselected random populations. Therefore, we believe that it is meaningful to show the characteristics of the population used in this study to readers along with the grain shape results.
- L115, you can use letters (such as P and B) instead of arrows.
Response: Thank you for your suggestion on that.
- In Table 1, delete the number 1. They are not necessary.
Response: All “the number1” have been removed from Table1.
- In Figure 2, correlation (r) between RLW and GL should be same but it is 0.68 in Table and 0.67 in Figure 2. Please check and correct one of them.
Response: In Figure 2, that was correctly revised as mentioned by the reviewer.
- In Figure 2, Change R to r
Response: In Figure 2, that was correctly revised as mentioned by the reviewer.
- L36-37, check LOD score?
Response: In L 136-137, we checked the LOD score. That was obtained by calculation of permutations in the program.
- L148, additive or additional?
Response: In 152, “additive” is correct. So, that was correctly revised in Table 2.
- L167-175, check paragraph.
Response: We reviewed the paragraph from 167 to 175 and found it to be acceptable.
- In Figure 4, red color could be light red color to read easily.
Response: Thank you for your suggestion on that. That was revised.
- L186, change r = -0.566 to r = -0.57
Response: L186 was changed following the reviewer’s comment.
- L197, explain P1 and P2.
Response: In 197, P1 and P2 were already explained at L192.
- Please insist on original name of parent cultivars or P1/P2.
Response: We mainly used name of parents and used the abbreviated form 'P1/P2' with the full name for clarity.
Discussion
- L218, GAO? Is it true?
Response: GAO was changed to Gao in the MS.
- L224, change QTLs to QTL
Response: The term 'QTLs' is correct. This means that it refers to the QTLs previously studied in GL.
- L227, please write this sentence at the end of sentence. “Linkage drags will not be problem in japonica rice breeding since QTL associated with GL was found to be alone on chromosome 2.
Response: There are no major hindrances in the original text that would prevent us from keeping the original sentence, which adequately explains the entire passage. We kindly request that the original sentence be maintained for better understanding.
- L256, change “Singh et al. 2015).” to (Singh et al. 2015).
Response: In L 256, that was correctly revised as mentioned by the reviewer.
M&M
- L334, explain or write full name of ICIM.
Response: There is full name of ICIM in introduction already at L104.
- L338, Data analysis to Data analyses because you performed more than one analysis.
Response: In L 256, that was correctly revised as mentioned by the reviewer.
- L342, write full name of DMRT for readers.
Response: There is full name of DMRT in introduction already at L156.
Table S1, S2 and S3
- Table S1, Change “Gene and Marker” to Genes and Makers, respectively.
Response: In Table S1, that was correctly revised as mentioned by the reviewer.
- Table S1, Genes should be italicized.
Response: In Table S1, that was correctly revised as mentioned by the reviewer.
- Table S2, some traits such as DH, CL, PL, and PN should be removed.
Response: We believe that data on traits are necessary for providing basic information on the selected promising lines.
- Table S2, change gamma (,) to dot (.).
Response: “,” changed correctly in Table S2.
- Table S2, three lines should be given in conclusion with important agronomical traits.
Response: We already mentioned the selcted three lines in the conclusion.
- Table S2, change agronomy to agronomical.
Response: In Table S2, that was correctly revised as mentioned by the reviewer.
- Table S2, change RGW to RLW. Is it true?
Response: In Table S2, that was correctly revised as mentioned by the reviewer.
I have read your mn with great pleasure.
Reviewer 2 Report
The manuscript “Application of a novel quantitative trait locus combination to improve grain shape without yield loss in rice (Oryza sativa L. 3 spp. japonica)” detected QTLs in grain shape and designed a QTL combination that increased grain length without penalty on yield. This manuscript could be under consideration for publish after the following revisions:
1. Only based on commercial preference to improve grain length is not enough. The advantage of grain length on cooking or taste should be described in introduction.
2. In figure 1, the phenotype didn’t represent as two years data.
3. In table 1, the number 1 on diagonal don’t offer meaningful information, just remove it.
4. Table 2, additional effect should be additive effect, it’s better to indicate which parent is the baseline.
5. Figure 3, the annotation label in the figure is easy to make confuse with multiple comparison rank label. And P < 0.001 usually is *** rather than **.
6. Figure 4, The TGW and RLW were not comparable, and putting the red and blue bar side by side is not appropriate.
7. Figure 5, The ty3 should have a fitting curve to demonstrate the combination effects of QTLs on TGW and RLW.
Author Response
Response to Reviewer 2 Comments
- Only based on commercial preference to improve grain length is not enough. The advantage of grain length on cooking or taste should be described in introduction.
Response: The reviewer's mentioned content has been included in the introduction (P1, Line 47).
- In figure 1, the phenotype didn’t represent as two years data.
Response: In Figure 1, we have clarified in the caption that we used the mean value of two years' data.
- In table 1, the number 1 on diagonal don’t offer meaningful information, just remove it.
Response: That was removed in the Table1.
- Table 2, additional effect should be additive effect, it’s better to indicate which parent is the baseline.
Response: That was revised correctly in the Table 2.
- Figure 3, the annotation label in the figure is easy to make confuse with multiple comparison rank label. And P < 0.001 usually is *** rather than **.
Response: That was revised correctly in the Figure 3.
- Figure 4, The TGW and RLW were not comparable, and putting the red and blue bar side by side is not appropriate.
Response: Comparisons were made only between the blue bars (TGW) or the red bars (RLW). The Duncan test was also used to analyze differences within the same color group (i.e., within the blue bars or within the red bars).
- Figure 5, The ty3 should have a fitting curve to demonstrate the combination effects of QTLs on TGW and RLW.
Response 2: Figure 5 displays scatterplots of the entire population for TGW and RLW, allowing us to examine the patterns of eight traits groups (ty1-ty8) and the parent. The results show which individuals satisfy both maternal TWG and paternal RLW, and in which group they belong. Therefore, we believe that additional graphs are not necessary and hope that you will take this into consideration.